# Selective Recovery of Palladium (II) from Metallurgical Wastewater Using Thiadiazole-Based Chloromethyl Polystyrene-Modified Adsorbent

**DOI:** 10.3390/ijms232012158

**Published:** 2022-10-12

**Authors:** Xiaoguo Zhang, Zhihong Chen, Zhaoneng Wan, Chali Liu, Renze He, Xiaoguang Xie, Zhangjie Huang

**Affiliations:** School of Chemical Science and Technology, Yunnan University, Kunming 650091, China

**Keywords:** adsorption, chloromethyl polystyrene, DFT calculations, palladium, separation

## Abstract

Selective adsorption of palladium from metallurgical wastewater containing Pt (IV), Rh (III), Ca^2+^, Cu^2+^, Fe^3+^, Ni^2+^, Pb^2+^, V^3+^, and Ti^4+^ has tremendous economic and environmental benefits. In this paper, a novel thiadiazole-based chloromethyl polystyrene-modified adsorbent, viz. 2, 5-bis-polystyrene-1,3,4-thiadiazole (PS-DMTD), was synthesized using chloromethyl polystyrene as the backbone. The experimental results show that PS-DMTD can selectively separate Pd (II) from metallurgical wastewater in a one-step adsorption process. The calculated saturation adsorption capacity of PS-DMTD for Pd (II) was 176.3 mg/g at 25 °C. The separation factors of *β*_Pd (II)/M_^n+^ (M^n+^: Pt (IV), Rh (III), Ca^2+^, Cu^2+^, Fe^3+^, Ni^2+^, Pb^2+^, V^3+^, and Ti^4+^) were all higher than 1 × 10^4^. FT-IR, XPS, and single-crystal X-ray diffraction showed that the adsorption of Pd (II) to PS-DMTD was primarily through a coordination mechanism. Density functional theory (DFT) calculations revealed that the other base metal ions could not coordinate with the PS-DMTD. Pt (IV) could not be adsorbed to PS-DMTD due to its strong chlorophilicity. Furthermore, Rh (III) existed as a polyhydrate, which inhibited Rh (III) diffusion toward the positively charged absorption sites on the PS-DMTD. These results highlight that PS-DMTD has broad application prospects in the recovery of Pd (II) from metallurgical wastewater.

## 1. Introduction

Palladium (Pd) is widely used in electronics, pharmaceuticals, the automotive industry, fuel cells, and other industries. It has excellent physical and chemical properties; for instance, a high melting point, good electrical conductivity, and high catalytic activity [1,2,3]. However, natural Pd reserves are inadequate to meet industrial needs [4,5]. In contrast, some metallurgical wastewaters contain large amounts of Pd [6,7], but Pd-containing wastewater has serious potential biological and environmental hazards that can cause kidney failure, asthma, etc. [8]. Therefore, more and more researchers are turning their attention to extracting Pd from metallurgical wastewater to obtain more economic and environmental benefits. However, other platinum group metals chemically similar to Pd (e.g., platinum, rhodium) are also widely present in metallurgical wastewater. Therefore, it is difficult to separate and recover Pd from metallurgical wastewater [9].

In recent decades, solvent extraction [10], chemical precipitation [11], membrane separation [12], ion exchange [13], and adsorption [14,15] have been widely applied to recover Pd from wastewater. Among these methods, adsorption offers the advantages of lower solvent consumption, no third-phase formation, and the ability to adsorb low-concentration solutions [16]. Therefore, it is regarded as one of the most promising approaches for the separation and recovery of Pd [17]. The selection of a suitable adsorbent is the most important aspect of Pd recovery through adsorption methods. So far, adsorbent materials such as activated carbon [18], impregnating resins [19], and metal–organic frameworks (MOFs) [20] have been used to adsorb and separate Pd. However, activated carbon shows poor selectivity and a low adsorption capacity for Pd. Hence, activated carbon is not suitable for separating Pd from metallurgical wastewater containing other metal ions. While impregnated resins often show high selectivity and high adsorption capacities for Pd, the bond strength between the extractant and the solid carrier typically is not a chemical bond, resulting in poor regeneration and reusability of the absorbent [21]. MOF-based adsorbents are considered next-generation materials based on their high surface area and multi-functionality. Yet, MOF-based adsorbents are unstable in acidic and basic media, and this low chemical stability limits the application of MOF-based adsorbents [22]. In summary, the current state of the absorbent materials is not ideal for the selective recovery of Pd from metallurgical wastewater.

Chloromethyl polystyrene (PS-Cl) is an ideal polymer for the creation of absorbents that are resistant to both strong acids and bases and are mechanically stable. Stable adsorbents are easily formed by replacing the reactive chlorine groups on the PS-Cl with active groups through a simple grafting reaction [23]. Due to their high stability, the chloromethyl polystyrene-modified adsorbents are ideal candidates for the adsorption and recovery of Pd [24,25,26]. However, the chloromethyl polystyrene-modified adsorbents also can adsorb other platinum group metal ions [27]. For instance, Kou et al. [28] prepared a series of chloromethyl polystyrene-modified adsorbents containing imidazole groups and studied the influence of different alkyl chains on the adsorption performance. In addition to Pd (II), the adsorbents also showed excellent adsorption performance for Pt (II)/(IV) and Rh (III). Until now, there are few studies on the relationship between the intrinsic structure and selectivity of chloromethyl polystyrene resin-modified adsorbents, which hinders their further practical applications. Moreover, highly selective adsorption and separation of Pd (II) from metallurgical wastewater through a one-step adsorption process with a chloromethyl polystyrene resin-modified adsorbent has not been reported in the literature.

In this paper, a novel chloromethyl polystyrene-modified resin adsorbent, 2,5-bis-polystyrene-1,3,4-thiadiazole (PS-DMTD), was prepared. PS-DMTD was used to selectively separate Pd (II) from metallurgical wastewater through a one-step adsorption process. The key adsorption parameters, including the optimum acidity, adsorption time, and initial concentration of PdCl_4_^2−^, were investigated. In addition, the adsorption behavior of PS-DMTD was analyzed using isotherms and kinetic models. The thermodynamic parameters of Pd (II) absorption with PS-DMTD were also systematically investigated. FT-IR, XPS, and single-crystal X-ray diffraction measurements were performed to explore the adsorption mechanism. Density functional theory (DFT) calculations of PS-DMTD further revealed the underlying mechanisms for the highly selective absorption of Pd (II).

## 2. Experiments

### 2.1. Reagents

Chloromethylated polystyrene (PS-Cl, 100-200 mesh) was purchased from Tianjin Nankai Hecheng Sci. & Tech. Co., Ltd. (Tianjin, China). Additionally, 2,5-dimercapto-1,3,4-thiadiazole (DMTD), hydrochloric acid, benzyl chloride, potassium iodide, dichloromethane, triethylamine, and methanol were purchased from Sigma-Aldrich Co., Ltd. (Saint Louis, MO, USA). PdCl_4_^2^^−^ (30 g/L) and metallurgical wastewater was provided by the Yunnan Mining Group Company (Kunming, China).

### 2.2. Synthesis of 2,5-bis-polystyrene-1,3,4-thiadiazole (PS-DMTD) and 2,5-bis(benzylthio)-1,3,4-thiadiazole (DTTD)

The novel chloromethyl polystyrene-modified resin adsorbent, PS-DMTD, was prepared as follows: To a 100 mL round bottom flask, 0.62 g PS-Cl, 1.05 g DMTD, 8 mL triethylamine, 16 mL methanol, and 0.7 g potassium iodide were added and stirred at 95 °C for 24 h. Then, ultrapure water, hydrochloric acid (HCl, 2.0 M) and methanol were used to wash the product. Finally, PS-DMTD was dried at 40 °C for 5 h.

The synthesis of 2,5-bis(benzylthio)-1,3,4-thiadiazole (DTTD) followed procedures in the literature [29]. The synthesis scheme for PS-DMTD and DTTD is shown in Appendix A. NMR and MS characterization results for DTTD are shown in Appendix A.

### 2.3. Adsorption Experiments

Unless otherwise stated, batch adsorption experiments were performed under the condition of pH 1.0. The effect of HCl concentration was studied as follows: 30 mg of PS-DMTD was added to 20 mL of Pd (II) solutions (500 mg/L) with an HCl concentration range of 0.01–2.0 M. After reaching adsorption equilibrium, the Pd (II) concentrations of the filtrates were determined via ICP-AES (Optima 8300, Perkin Elmer, Waltham, MA, USA).

To measure the adsorption kinetics, 30 mg of PS-DMTD was placed in 20 mL of Pd (II) solution (500 mg/L). After being oscillated (25 °C, 230 rpm) for different time intervals, the concentration of Pd (II) was measured.

To measure the adsorption isotherms, 30 mg of PS-DMTD was placed in a series of solutions containing different concentrations of Pd (II) (40–1300 mg/L, 20 mL total volume). After being shaken (25 °C, 230 rpm) for 240 min, the adsorption capacity (*q_e_*) of the PS-DMTD for Pd (II) was calculated according to Equation (1).
(1)$$q_{e}=\frac{V\left(C_{0}-C_{e}\right)\;}{m}$$
where *m* (g) is the mass of PS-DMTD, *C*_0_ (mg/L) and *Ce* (mg/L) are the concentrations of Pd (II) before and after adsorption, respectively, and *V* (L) is the volume of the PdCl_4_^2^^−^ solution.

To determine the thermodynamic parameters of the absorption, 30 mg of PS-DMTD was placed in 20 mL of a solution containing 500 mg/L Pd (II). The sample vials were shaken (230 rpm) at different temperatures. After being shaken for 240 min, the concentration of Pd (II) in the filtrate was measured using ICP-AES. Each adsorption experiment was repeated three times, and the final error was ≤3%.

### 2.4. Physical and Chemical Analysis

The elemental contents of C, H, and N in PS-DMTD were measured through elemental analysis (Vario EL III, Germany). Infrared spectroscopy (FT-IR, NICOLET 8700, Thermo Fisher Scientific, Waltham, MA, USA) was used to determine the functional groups present in the samples. An X-ray photoelectron spectrometer (XPS, K-Alpha^+^, Thermo Fisher Scientific, Waltham, MA, USA) was used to determine the chemical states of the different elements in the PS-DMTD. The morphology of the PS-DMTD absorbent was observed using scanning electron microscopy (SEM-EDS, Nova NanoSEM 450, FEI Co., Ltd. Portland, OR, USA). A thermogravimetric analysis instrument (TGA, TA SDT-Q600, TA Instruments, New Castle, DE, USA) was used to examine the thermal stability of the materials, and ^1^H NMR spectra (Avance500, Bruker, Germany) were measured to analyze the molecular structure of the synthesized DTTD product. The crystal structure of the Pd (II)-DTTD adduct was determined via single-crystal X-ray diffraction (SMART APEX II CCD, Bruker Corporation, Karlsruhe, Germany).

## 3. Results and Discussion

### 3.1. Characterization of PS-DMTD

Appendix A shows the elemental contents of C, H, and N in PS-Cl and PS-DMTD. Notably, the N content in PS-DMTD was significantly higher than that of PS-Cl, and the C and H contents were lower than those of PS-Cl due to the replacement of Cl with DMTD in the PS-DMTD [30]. The elemental analysis results confirmed that DMTD was successfully introduced onto the PS-Cl.

The FTIR spectra of PS-Cl, DMTD, PS-DMTD, and PS-DMTD-Pd (II) are shown in Figure 1. The characteristic bands located at 677 cm^−1^ and 1264 cm^−1^ in the spectrum of PS-Cl (Figure 1a) corresponded to the stretching vibrations of C-Cl and the bending vibrations of CH_2_-Cl, respectively [24]. The characteristic peak at 2478 cm^−1^ found in the spectrum of DMTD (Figure 1b) corresponded to the stretching vibrations of -SH [31]. Comparing the spectra for DMTD and PS-DMTD showed that the peak at 2478 cm^−1^ disappeared in the spectrum of PS-DMTD (Figure 1c), and new peaks appeared at 1055 cm^−1^ and 1605 cm^−1^, which were assigned to the stretching vibrations of C-S [32] and the stretching vibrations of C=N [33], respectively. The appearance of these new peaks supported that the Cl groups in PS-Cl were replaced by DMTD. After the adsorption of Pd (II), the peaks of both C=N and C-S shifted to high frequencies (4 cm^−1^ and 7 cm^−1^, respectively) (Figure 1d), suggesting that the N and S atoms were involved in the adsorption of Pd (II) [25].

The XPS spectra were analyzed to better understand the Pd (II) adsorption mechanism, and the results are shown in Figure 2a. Bands corresponding to N 1s, S 2s, and S 2p were seen in the XPS spectrum measured for PS-DMTD. In addition, the Cl 2p band in the spectrum of PS-DMTD was significantly weaker than that of PS-Cl, indicating that most of the Cl groups in PS-Cl were replaced by DMTD. Furthermore, the Pd 3d peak was present in the PS-DMTD-Pd (II) spectrum, which showed that Pd (II) was successfully adsorbed onto PS-DMTD. The high-resolution N 1s XPS spectra before and after adsorption, are shown in Figure 2b. Before the adsorption of Pd (II), two peaks were found at 399.84 eV and 400.96 eV, belonging to N-N [29] and C=N [32], respectively. After Pd (II) adsorption, the N-N peak shifted to a binding energy (BE) of 399.94 eV, and a new peak appeared at a BE of 402.51 eV which was ascribed to the formation of N-Pd bonds [34]. As shown in Figure 2c, the S 2p spectra could be deconvoluted into two components attributed to C-S-C (thioether, 163.54 eV) and C-S-C (thiophene, 164.94 eV) [35]. After the adsorption, the BE of the C-S-C (thioether) and C-S-C (thiophene) shifted to 163.84 eV and 165.22 eV, respectively. In addition, a new peak assigned to S-Pd bonds appeared at 162.44 eV [36]. The peaks at 337.43 eV and 342.67 eV were ascribed to the Pd-Cl bonds in PdCl_4_^2−^, and the doublet peaks (BE = 338.21 eV, 343.36 eV) likely corresponded to the binding energies of Pd-N or Pd-S bonds (Figure 2d) [37,38,39]. The XPS results further confirmed that DMTD was successfully grafted onto the PS-Cl and also indicated that Pd (II) coordinated with the N and S atoms in the PS-DMTD absorbent.

The N_2_ adsorption–desorption isotherms of PS-Cl and PS-DMTD are shown in Figure 3a, and the isotherm was similar to the typical IV adsorption isotherms. Figure 3b shows that the corresponding pore size distributions were indicative of a mesoporous structure and were advantageous for adsorption. PS-DMTD had a specific surface area and total pore volume of 364.03 m^2^/g and 1.54 cm^3^/g, respectively, which were smaller than that of PS-Cl (Table 1) due to the DMTD covering the surface of the synthesized PS-DMTD [40].

SEM images showed that the PS-DMTD were spherical in shape with diameters ranging from 80–150 μm (Figure 4a,b). Figure 4c shows the corresponding maps for PS-DMTD-Pd (II). The images show that N and S were uniformly distributed on the surface of PS-DMTD, which can be attributed to the dense and thin thiazole layer grafted on the surface of the particles after the reaction [41]. Notably, Pd (II) was also uniformly distributed on the surface of PS-DMTD-Pd (II), which further highlights that PS-DMTD effectively adsorbed Pd (II).

Appendix A shows the TGA curves of PS-Cl and PS-DMTD. During thermal degradation, PS-Cl showed two weight loss peaks. The first weight loss peak occurred near 146 °C, and the second decomposition heat signal appeared at approximately 442 °C. In contrast, the TGA curve for PS-DMTD only showed one thermal degradation process at 437 °C. At 900 °C, the residues of PS-Cl and PS-DMTD were about 13.3% and 21.0%, respectively. The TGA results revealed that the PS-DMTD was more thermally stable than PS-Cl due to the formation of stable covalent bonds between PS-Cl and DMTD during the synthesis [42].

### 3.2. The Effect of HCl Concentration

According to Appendix A, the Pd (II) adsorption capacity remained almost constant for HCl concentrations between 0.01–0.1 M. However, the degree of protonation on the PS-DMTD absorbent increased significantly as the HCl concentration increased from 0.1 to 2.0 M, resulting in a decrease in the number of active sites involved in coordination with Pd (II) and a decrease in the absorption capacity [1]. When the concentration of HCl is lower than 0.01 M, Pd (II) is readily hydrolyzed [25]. Therefore, the optimal pH range of PS-DMTD adsorption of Pd (II) is 1.0 to 2.0. Unless otherwise stated, the following experiments were performed under the condition of pH 1.0.

### 3.3. Adsorption Kinetics Studies

Appendix A shows the relationship between contact time and adsorption capacity. The adsorption capacity of PS-DMTD for Pd (II) increased rapidly for 80 min and reached equilibrium at 240 min. The experimental data were fitted with pseudo-first-order and pseudo-second-order kinetic models, as shown in Equations (2) and (3) [43].
(2)$$log(q_{e}-q_{t})=logq_{e}-\frac{k_{1}t}{2.303}$$
(3)$$\frac{t}{q_{t}}=\frac{1}{k_{2}\cdot q_{e}^{2}}+\frac{t}{q_{e}}$$
where *q_e_* (mg/g) is the adsorption capacity of PS-DMTD for Pd (II) at equilibrium, and *q_t_* (mg/g) is the adsorption capacity of PS-DMTD for Pd (II) at t time. *k_1_* (min^−1^) and *k*_2_ (g·mmol^−1^·min^−1^) are the rate constants.

Table 2 shows the parameters fitted by the kinetic models, and fit curves are shown in Appendix A. Table 2 and Appendix A show that the process of Pd (II) adsorption onto PS-DMTD was mainly through chemical adsorption.

### 3.4. Adsorption Isotherm Studies

The experimental data were non-linearly fitted with the Langmuir and Freundlich models, as shown in Equations (4) and (5) [25].
(4)$$\;q_{e}=\frac{K_{L}C_{e}q_{m}}{1+K_{L}C_{e}}$$
(5)$$q_{e}=K_{F}C_{e}^{\frac{1}{n}}$$
where *q_m_* (mg/L) is the maximum adsorption capacity, *K_L_* is the Langmuir adsorption constant, and *K_F_* and *n* are the Freundlich adsorption constants. Figure 5 shows the fitted curves for the two models, and Table 3 shows the corresponding fitted parameters. The computing method of the adjusted determination coefficient (R^2^_adj_), the residual sum of squares (RSS), and the Bayesian information criterion (BIC) are shown in Equations (S1)–(S3) [44,45].

Table 3 shows that the R^2^_adj_ value of the Langmuir isotherm model is higher than that of the Freundlich model. On the contrary, the RSS and BIC values of the Langmuir isotherm model are lower than those of the Freundlich model. Hence, the Langmuir model is better fitted for Pd (II) adsorption onto PS-DMTD [46,47,48]. In addition, the calculated Langmuir adsorption capacity of PS-DMTD for Pd (II) was 176.3 mg/g at 25 °C.

### 3.5. Adsorption Thermodynamics for Pd (II)

The adsorption performance of PS-DMTD for Pd (II) at different temperatures was investigated. The thermodynamic equilibrium constant *K_d_*, the entropy changes Δ*S*, the standard enthalpy changes Δ*H*, and the free energy change Δ*G*, were calculated using Equations (6)–(8):(6)$$K_{d}=\frac{(C_{0}-C_{e})V}{mC_{e}}$$
(7)$$LnK_{d}=-\frac{\Delta H}{RT}+\frac{\Delta S}{R}$$
(8)ΔG=ΔH−TΔS

Appendix A shows the relationship between ln *K_d_* and 1/*T*, and Table 4 shows the values of Δ*H* and Δ*S.* Here, Δ*H* > 0 and Δ*G* < 0, which indicated that the adsorption was an endothermic reaction and spontaneous, respectively [49]. Δ*S* > 0 suggested that the randomness of the interface between the PS-DMTD and PdCl_4_^2−^ increases and the high temperature is beneficial to adsorption. The absolute value of *T*Δ*S* > Δ*H* indicated that the entropy change was the main factor influencing the adsorption process [50].

### 3.6. Application in Metallurgical Wastewater

To investigate the practicality of PS-DMTD, it was used to adsorb Pd (II) from metallurgical wastewater. The composition of the metallurgical wastewater is shown in Appendix A. The distribution ratio (*D*), adsorption efficiency (*E*), and separation factor (*β*) were obtained using Equations (9)–(11). Figure 6 shows the adsorption efficiency (*E*) and the distribution ratio (*D*) of PS-DMTD for the different metal ions in the metallurgical wastewater. The result indicated that the separation factors (*β*) of Pd (II) and Pt (IV), Rh (III), Ca^2+^, Cu^2+^, Fe^3+^, Ni^2+^, Pb^2+^, V^3+^, and Ti^4+^ were 1.3 × 10^4^, 2.3 × 10^4^, 2.3 × 10^4^, 2.6 × 10^4^, 1.7 × 10^5^, 3.3 × 10^4^, 1.9 × 10^4^, 6.7 × 10^4^, and 1.0 × 10^5^, respectively. Thus, PS-DMTD was highly selective towards separating Pd (II) from the metallurgical wastewater.
(9)$$D=\frac{V\left(C_{0}-C_{e}\right)}{mC_{e}}$$
(10)$$\beta =\frac{D_{Pd\;\left(II\right)}}{D_{M^{n+}}}$$
(11)$$E=\frac{C_{0}-C_{e}}{C_{0}}\times 100\%$$
where *D_Pd_
*_(*II*)_ is the distribution ratio of Pd (II) and *D_Mn+_* is the distribution ratio of Pt (IV), Rh (III), Ca^2+^, Cu^2+^, Fe^3+^, Ni^2+^, Pb^2+^, V^3+^, and Ti^4+^.

### 3.7. Regeneration and Recovery

The circulation performance of an adsorbent is an important indicator. The PS-DMTD loaded with Pd (II) was treated with 20 mL of the acidified thiourea solution to desorb the Pd (II). Then, the PS-DMTD was washed successively with anhydrous ethanol and ultrapure water, and finally dried under vacuum at 40 °C for 12 h. The recovery efficiency of Pd (II) was still over 92.0% after five cycles (Figure 7), implying that PS-DMTD has an excellent regeneration and reuse performance.

## 4. Adsorption Mechanism

### 4.1. DFT Results

To analyze and reveal the selectivity and mechanism of anion metal chlorides adsorbed by the PS-DMTD, one structural unit of the PS-DMTD was chosen as a model molecule. The model molecule is denoted by L, viz. DTTD. Its optimized structure and Mulliken atomic charges are shown in Figure 8. Protonation of PS-DMTD, the chlorophilic property of metal ions, the coordination strength of the model molecule L with the anionic metal chloride, and the average charge density of anionic metal chlorides were investigated using density functional theory (DFT) calculations.

#### 4.1.1. Protonation of PS-DMTD

Under acidic conditions, the N atom of the PS-DMTD may be protonated and PS-DMTD may adsorb metal chloride anions as an anion-exchange resin. Thermodynamic parameters of the following protonation reactions of L were calculated at the B3LYP/6-311++G**/SDD//B3LYP/6-31G**/SDD level. Table 5 shows the calculation results; Figure 9 gives the optimized structure of L protonated at the N atom and its Mulliken atomic charges.
(12)$$H_{3}O^{+}+N\left(L\right)⇌HN{\left(L\right)}^{+}+H_{2}O$$
(13)$$H_{3}O^{+}+NH\left(L\right)\to N_{2}H_{2}\left(L\right){+H}_{2}O$$

From the values of the thermodynamic parameters for protonation reactions on the N atom (Table 5), it is expected that the PS-DMTD would be highly protonated in an acidic solution. The protonation of the first N atom is a highly exothermic process. In contrast, the protonation of the second N atom is endothermic (43.43 kcal/mol), and one of the two N atoms is easier protonated. Furthermore, protonation of the S atom should not occur because the S atom of L has a positive charge of 0.059 (Figure 8). The protonated adsorption active site suggests that PS-DMTD may act as an anion-exchange resin. Therefore, the adsorption of metal ions by PS-DMTD in acidic solutions may involve two adsorption mechanisms: one is the coordination mechanism of N and S atoms (as ligand atoms of L) to metal ions, and the other is the ionic association mechanism in which the metal chloride anions are absorbed by PS-DMTD (as an anion-exchange resin).

It is well-known that base metal ions (Ca^2+^, Cu^2+^, Fe^3+^, Ni^2+^, Pb^2+^, V^3+^, and Ti^4+^) exist mainly in their cationic forms in an aqueous solution. Thus, Ca^2+^, Cu^2+^, Fe^3+^, Ni^2+^, Pb^2+^, V^3+^, and Ti^4+^ were not adsorbed by PS-DMTD as an anion-exchange resin via an ionic association mechanism. On the other hand, the highly protonated adsorption active site is positively charged and repels base metal cation diffusion and prevents base metal cations from getting close to the active adsorption site. Furthermore, the S atom (as a possible coordination atom) is also positively charged at 0.145 and 0.128 (Figure 9) after the protonation of PS-DMTD. Hence, positively charged base metal ions cannot coordinate with the PS-DMTD via a coordination mechanism. This is in full agreement with the experimental results that showed PS-DMTD had no adsorption capacity for base metal cations.

#### 4.1.2. Chlorophilic Property of Metal Cations

In hydrochloric acid media, Pd (II) and Pt (IV) mainly exist in the forms of metal chloride anions PdCl_4_^2−^ and PtCl_6_^2−^, respectively. Whereas for Rh (III), RhCl_6_^3−^ is unstable, easily hydrated, and mainly exists as polyhydrate ions in acidic solution [51,52]. The instability of RhCl_6_^3-^ and the formation of polyhydrates inhibits the diffusion of Rh (III) towards the absorption sites, which is why Rh (III) was hardly absorbed by PS-DMTD. To verify this observation, we calculated the reaction energies of water molecules substituting the chlorine anion of RhCl_6_^3−^ at the B3LYP/def2-QZVP level. The calculation results show that the substitution process of the first chlorine anion by a water molecule is highly exothermic (137.75 kcal/mol), and the second chlorine anion substituted by a water molecule is even more exothermic (191.92 kcal/mol). This indicates that RhCl_6_^3−^ is easily hydrated and forms polyhydrate species in an aqueous solution.

In theory, the adsorption processes Pd (II) and Pt (IV) may involve two mechanisms, i.e., the coordination mechanism and the anion-exchange mechanism. In terms of the coordinating procedure, the metal chloride anion first diffuses from the bulk solution to near the protonated adsorption active site. After the deprotonation of the N atom, the N and S coordinating atoms in PS-DMTD then substitute one or two chloride ions (Cl^−^) and coordinate with the metal ion. Thus, coordinating interactions should overcome both the energy barrier of the deprotonation of the N atom and the energy barrier for substituting one or two chloride ion (Cl^−^) ligands in the metal chloride anion. Thus, the coordination strength of Cl^−^ with a metal ion in the metal chloride anion (in other words, the chlorophilic property of metal ions) is an important aspect that impacts the adsorption of metal chloride anions via the coordination mechanism. The average coordination bonding energy (ACE) of the metal chloride anions, PdCl_4_^2−^, RhCl_6_^3−^, and PtCl_6_^2−^, were calculated at the B3LYP/def2-QZVP level according to the dissociation reaction (Equation (14)), as shown in Table 6.
(14)$${\left[MCl_{x}\right]}^{z-}\to M^{\left(x-z\right)+}+xCl^{-}\;$$

As seen in Table 6, the ACE value of PtCl_6_^2−^ is 405.24 kcal/mol and is much larger than that of the other two metal chloride anions, and this indicates that the Pt (IV) has the strongest chlorophilic property and the ACE value of PdCl_4_^2−^ is the smallest and has the weakest chlorophilic property. The calculation results show that PdCl_4_^2−^ is the most likely to be adsorbed via the coordination mechanism and PtCl_6_^2−^ is the most unlikely to be adsorbed via the coordination mechanism.

#### 4.1.3. Coordination Strength of PS-DMTD with Metal Ions

Another important factor that impacts the adsorption of metal chloride anions via the coordination mechanism is the coordination strength of PS-DMTD with metal ions. Thus, we calculated the reaction energy changes (DE) in the N and/or S atoms of L by substituting one or two chloride ion (Cl^−^) ligands of [MCl_x_]^z−^ to coordinate with the metal ion (Equations (15) and (16)). The energy value (CE) was calculated according to Equation (17), which represents the coordination strength of PS-DMTD with metal ions. The calculation results of some typical coordination models (Figure 10) are listed in Table 7, and the ACE values of metal chloride anions are also listed for convenient comparison.
(15)$$L+[MCl_{x}{]}^{z-}\;\to [LMCl_{x-y}{]}^{\left(z-y\right)-}+yCl^{-}$$
(16)$$\;DE=E_{L}+E_{{\left[MCl_{x}\right]}^{z-}}-E_{{\left[LMCl_{y}\right]}^{\left(z-y\right)-}}-yE_{Cl^{-}}$$
(17)$$\;CE=DE+yACE_{{\left[MCl_{x}\right]}^{z-}}$$

Both the N and S atoms of L can simultaneously coordinate with metal ions (Figure 10), and the PS-DMTD can coordinate with the metal ions. One adsorption site of L can coordinate with one or two metal chloride anions. In terms of the coordination strength values (CE) in Table 7, the coordination interaction of Pt (IV) with L is the strongest, and that of Pd (II) and Rh (III) are comparable. However, regarding the coordination of L with the N and S atoms, substituting one or two Cl^-^ of Pd (II) and Rh (III)-based chloride anions is energetically favored, whereas the coordination of L substituting a chloride ion of PtCl_6_^2−^ is endothermic (the DE values are shown in Table 7). This suggests that the coordination of L with Pd (II) and Rh (III) is stronger than that of the chloridion ligand and suggests that PS-DMTD is in favor of absorbing the Pd (II) ion with the coordination mechanism. On the other hand, Pt (IV) cannot be efficiently absorbed by the PS-DMTD resin via the coordination mechanism due to its strong chlorophilicity. Although the coordination strength of L with Rh (III) is strong enough, it hardly shows an absorption capacity, indicating that Rh (III) mainly exists as a polyhydrate, as a neutral complex (RhCl_3_(H_2_O)_3_) and/or as positive ions in 0.1M HCl media [52], which inhibits its diffusion toward the positively charged absorption site of PS-DMTD.

As mentioned above, the DE values of Pd (II) are positive (Table 7), suggesting that substitution coordination reactions of PdCl_4_^2−^ with L are exothermic. However, Pd (II) adsorption is endothermic, with an experimental value of 36.53 kJ/mol. This is because the deprotonation process of the protonated PS-DMTD occurs before the substitution coordination reaction of PS-DMTD with PdCl_4_^2−^, and the deprotonation of the protonated PS-DMTD is a highly endothermic process (Table 5). This also provides evidence that the adsorption of Pd (II) is via the coordination mechanism instead of via the anion-exchange mechanism since the deprotonation of the protonated PS-DMTD occurs. The crystallization of the Pd (II)-DTTD adduct (CCDC: 2165849) was also obtained by mixing DTTD with the PdCl_4_^2−^ solution. Figure 11 shows the crystal structure and the projection of the crystal along the a-axis, and the crystal data are shown in Appendix A. The result indicates that the single crystal formed by Pd (II), two Cl atoms, and two DTTDs is a zero-dimensional structure and demonstrates a strong coordination ability between N and Pd (II).

#### 4.1.4. Low Absorption via an Anion-Exchange Mechanism

The absorption site of PS-DMTD is partly positively charged and may absorb metal chloride anions as an anion-exchange resin in theory; the ionic association mechanism may also contribute to the absorption capacity of Pd (II). Herein, we will discuss these hypotheses based on the experimental and theoretical calculation results.

Adsorption via the anion exchange is expected to follow trends with the “smallest charge density” [1,53], i.e., adsorption through the anion-exchange mechanism favors metal chloride anions with a smaller charge density because of the loose water solvation. We calculated the volumes of the metal chloride anions and their average charge densities, i.e., the anion’s charge divided by its volume. The results are listed in Table 8.

Based on expected trends with charge densities, it can be concluded that PtCl_6_^2−^ would be more easily adsorbed than PdCl_4_^2−^ via an anion-exchange mechanism because PtCl_6_^2−^ has a smaller charge density. However, the experimental separation factor (*β*_Pd (II)/Pt (IV)_) is 1.3 × 10^4^, which means that PS-DMTD is highly selective towards the absorption of Pd (II). In contrast, Pt (IV) is hardly absorbed via the anion-exchange mechanism. Hence, the adsorption of Pd (II) is via a coordination mechanism and the anion-exchange mechanism hardly contributes to the absorption capacity of Pd (II). This is because the degree of protonation of the PS-DMTD resin is not enough to be used as an anion-exchange resin, and Pd (II) is absorbed via the coordination mechanism.

## 5. Conclusions

In this study, a new modified chloromethyl polystyrene resin adsorbent, PS-DMTD, was successfully synthesized by grafting 2,5-dimercapto-1,3,4-thiadiazole onto PS-Cl. PS-DMTD could adsorb and separate Pd (II) from the metallurgical wastewater through a one-step adsorption process. The adsorption process is in accordance with the pseudo-second-order kinetic and Langmuir model. The highly selective absorption of Pd (II) was discussed in detail based on the spectral characterization and experimental results. DFT calculations were used to further reveal the adsorption mechanism at the molecular level. After protonation, the PS-DMTD resin showed a positively charge, which prevented the base metal cations from approaching the active adsorption site. Therefore, the base metal cations were not absorbed by the PS-DMTD in the metallurgical wastewater. Pd (II) was most likely to be adsorbed via a coordination mechanism, and the degree of protonation of PS-DMTD was not sufficient to absorb Pt (IV) in 0.1 M HCl media. Furthermore, Rh (III) mainly exists as polyhydrate ions in this acidity, which inhibited its diffusion toward the positively charged absorption site of PS-DMTD. Hence, PS-DMTD was highly selective in the separation and recovery of Pd (II) from metallurgical wastewater in a one-step adsorption process.

## Figures and Tables

**Figure 1 ijms-23-12158-f001:**
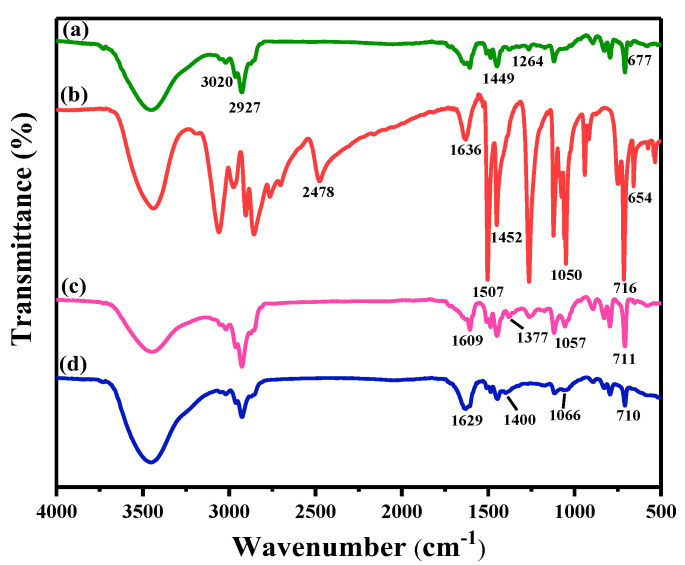
FTIR spectra of (**a**) PS-Cl, (**b**) DMTD, (**c**) PS-DMTD, and (**d**) PS-DMTD-Pd (II).

**Figure 2 ijms-23-12158-f002:**
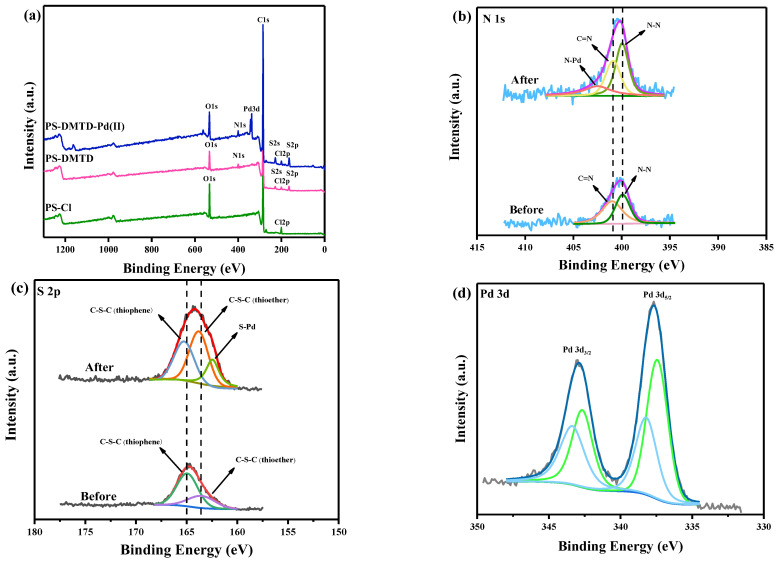
XPS spectra analysis: (**a**) full spectra of PS-Cl, PS-DMTD and PS-DMTD-Pd (II), (**b**) N 1s, (**c**) S 2p, and (**d**) Pd 3d.

**Figure 3 ijms-23-12158-f003:**
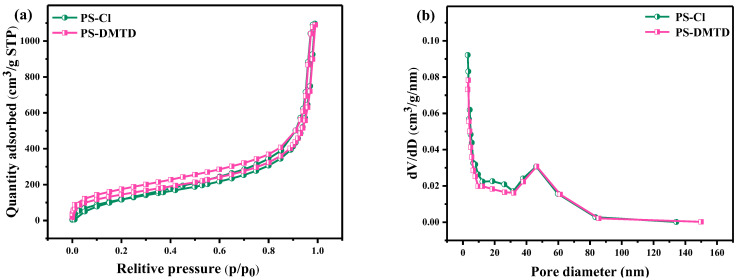
(**a**) N_2_ adsorption–desorption isotherms for PS-Cl and PS-DMTD and (**b**) the corresponding pore size distribution (BJH).

**Figure 4 ijms-23-12158-f004:**
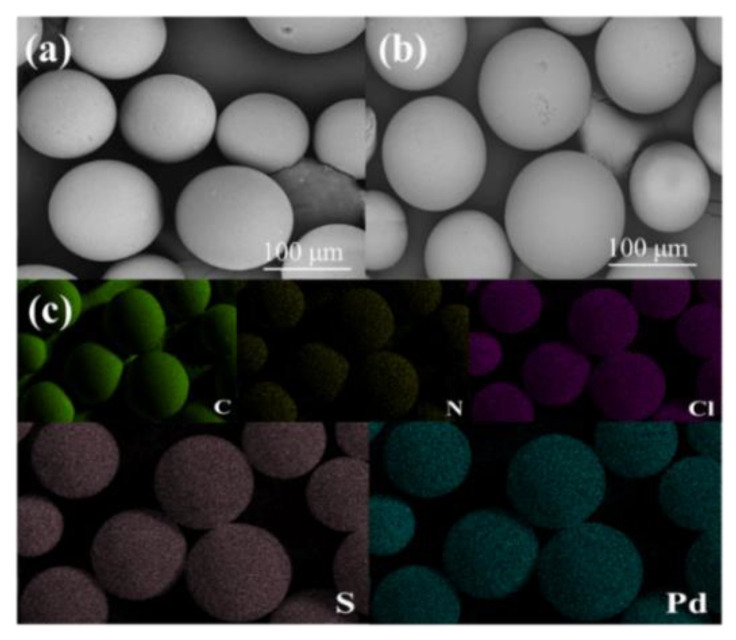
(**a**) SEM image of PS-DMTD, (**b**) PS-DMTD-Pd (II), and (**c**) the elemental mapping images of PS-DMTD-Pd (II).

**Figure 5 ijms-23-12158-f005:**
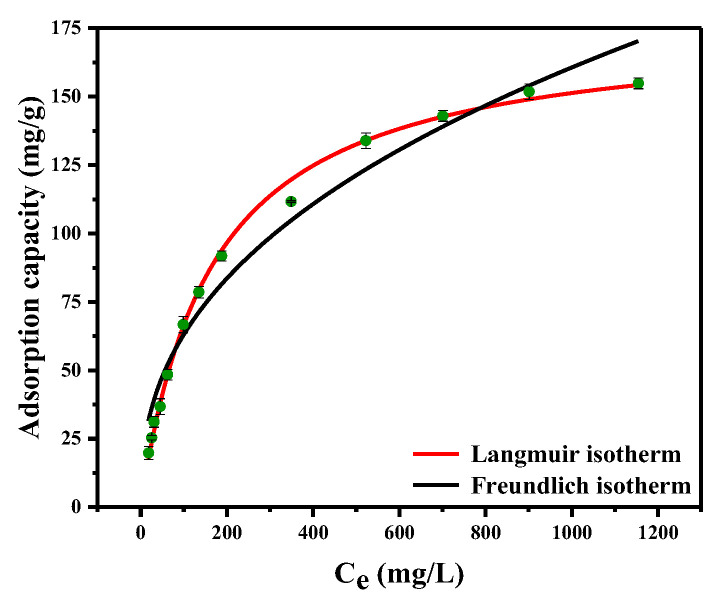
The non-linearly fitted curves of Langmuir and Freundlich model.

**Figure 6 ijms-23-12158-f006:**
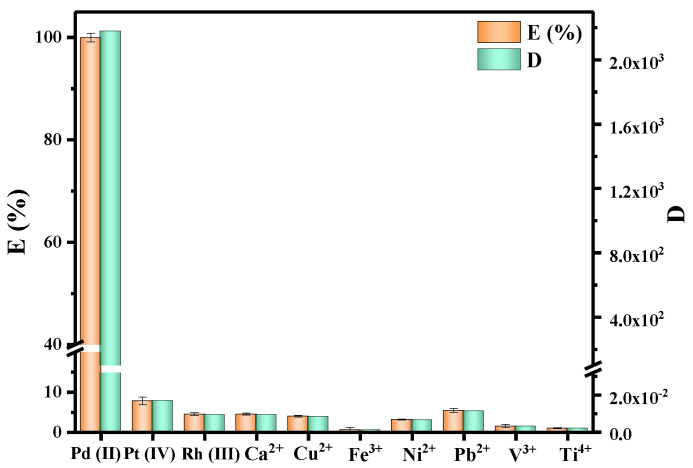
Separation of Pd (II) from the metallurgical wastewater. m_PS-DMTD_ = 0.1 g, C_Pd (II)_ = 109 mg/L, pH = 1.0, T = 298.15 K, t = 240 min, and V = 0.02 L.

**Figure 7 ijms-23-12158-f007:**
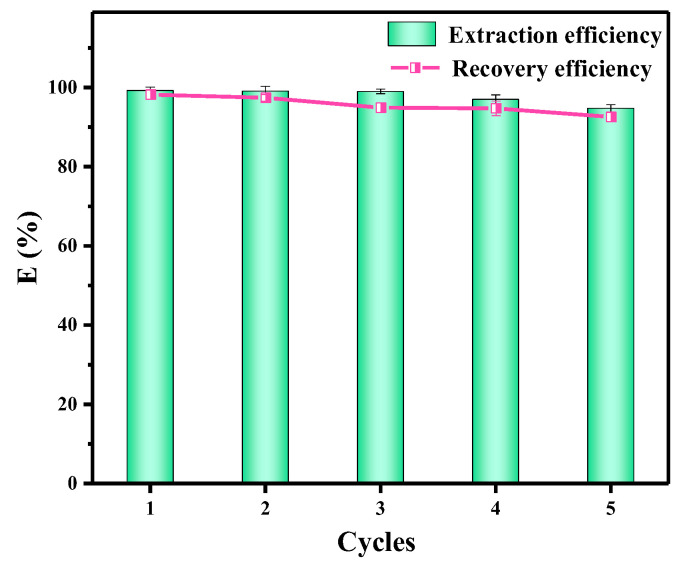
Adsorption of Pd (II) by PS-DMTD in the adsorption–desorption cycle. m_PS-DMTD_ = 0.040 g, C_Pd (II)_ = 100 mg/L, pH = 1.0, T = 298.15 K, t = 240 min, and V = 0.02 L.

**Figure 8 ijms-23-12158-f008:**
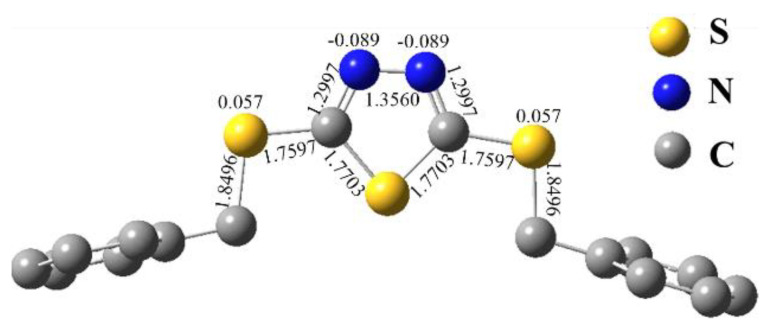
The optimized structure and Mulliken atomic charge of the model molecule L (at B3LYP/6-31G**/SDD level).

**Figure 9 ijms-23-12158-f009:**
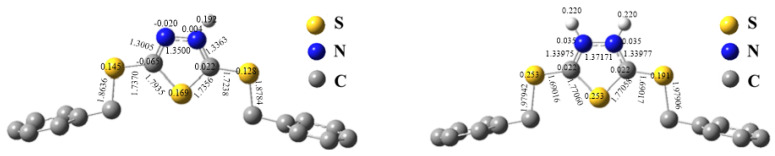
The optimized structure and Mulliken atomic charge of the protonated model molecule of PS-DMTD (at B3LYP/6-31G**/SDD level).

**Figure 10 ijms-23-12158-f010:**
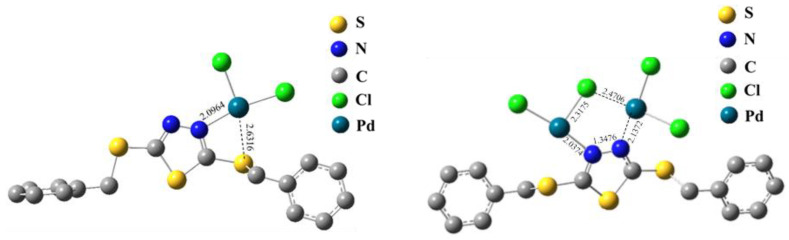
The optimized structure of two typical coordination complexes (at B3LYP/def2-QZVP level).

**Figure 11 ijms-23-12158-f011:**
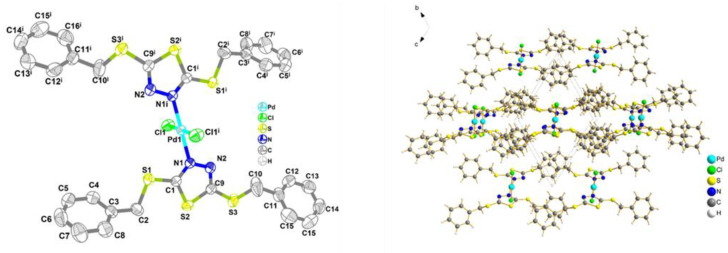
Crystal structure and the corresponding projection along the a-axis.

**Table 1 ijms-23-12158-t001:** BET data of PS-Cl and PS-DMTD.

	Specific Surface Area m^2^/g	Total Pore Volume cm^3^/g
PS-Cl	417.65	1.65
PS-DMTD	364.03	1.54

**Table 2 ijms-23-12158-t002:** Kinetic data of adsorbed Pd (II).

	Pseudo-First-Order Kinetic Model		Pseudo-Second-Order Kinetic Model	
	*k*_1_ (min^−1^)	*R* ^2^	*k*_2_ (g·mmol^−1^·min^−1^)	*R* ^2^
PS-DMTD	0.0205	0.8737	0.00143	0.9987

**Table 3 ijms-23-12158-t003:** Non-linear adsorption parameters of PS-DMTD for Pd (II).

Isotherm Models	Parameters	
Non-linear Langmuir model	*q_m_*_, *cal*_ (mg/g)	176.3
	*q_m_*_, *exp*_ (mg/g)	155.2
	*K_L_* (L/mg)	0.0061
	R^2^_adj_	0.9962
	RSS	105.10
	BIC	16.93
Non-linear Freundlich model	*K_F_* (mg L^1/n^ g^−1^ mg^−1/n^)	9.724
	n^−1^	0.4060
	R^2^_adj_	0.9648
	RSS	985.95
	BIC	21.36

**Table 4 ijms-23-12158-t004:** The values of Δ*H*, Δ*S*, and Δ*G*.

	T	Δ*G*	Δ*H*	Δ*S*
	(K)	(kJ·mol^−1^)	(kJ·mol^−1^)	(J·mol^−1^·K^−1^)
PS-DMTD	293.15	−14.84	36.53	175.22
	298.15	−15.71		
	303.15	−16.59		
	308.15	−17.47		
	313.15	−18.34		

**Table 5 ijms-23-12158-t005:** Thermodynamic parameters of protonation of L (at standard atmosphere, 298.15 K, in gas phase).

Parameters	Δ*G* (kcal/mol)	Δ*H* (kcal/mol)	Δ*S* (Cal/T·mol)
Reaction (1)	−59.59	−59.81	−0.37
Reaction (2)	40.35	43.43	5.17

**Table 6 ijms-23-12158-t006:** The average coordination energy (in kcal/mol) of metal cation with chloridion in complex [MClx]-z (at B3LYP/def2-QZVP level).

Anion	PdCl_4_^2−^	RhCl_6_^3−^	PtCl_6_^2−^
ACE	172.24	211.29	405.24

**Table 7 ijms-23-12158-t007:** The energy change (DE) in some coordination reactions of L with [MClx]z^−^ and coordination strength (CE) of L with metal chloride anion (at B3LYP/def2-QZVP level).

Coordination Complex	DE	CE	ACE
	(kcal·mol^−1^)	(kcal·mol^−1^)	(kcal·mol^−1^)
[PdCl_2_]-L	6.51	350.99	344.48
[Pd_2_Cl_4_]^−^-L	13.33	351.14	344.48
[RhCl_4_]^−^-L	0.41	422.99	422.58
[RhCl_4_]^−^·H_2_O-L	52.14	263.42	211.29
[PtCl_4_]^−^-L	−10.51	799.97	810.48
[Pt_2_Cl_8_]-L	−51.20	784.88	810.48

**Table 8 ijms-23-12158-t008:** The data of metal chloride anion (at B3LYP/6-31G**/SDD level).

Species	Molecular Volume (Å^−3^)	Average Charge Density (10^−3^e. Å^−3^)
PtCl_6_^2−^	225.440	8.87
PdCl_4_^2−^	178.460	11.21

## Data Availability

Not applicable.

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
