# Peer review of "Selective Recovery of Palladium (II) from Metallurgical Wastewater Using Thiadiazole-Based Chloromethyl Polystyrene-Modified Adsorbent"

_ijms, 2022, doi:10.3390/ijms232012158_

Round 1
Reviewer 1 Report
1. Abstract section is well written. However, results have not shown numerical values. Also selectivity for Pd ion has been claimed but which other ions were present in the solution has not shown.
2. Arrange the keywords in alphabetical order. Also omit those words that are part of manuscript title.
3. Introduction is too long that needs to be revised.
4. A heading or sentence starts with a capital word. Please revise 2.1 section heading
5. Many abbreviations are used but not defined at all
6. Line 119, please remove space before K symbol
7. Line 120, a sentence does not start from a numerical value, use about or approximately before
8. The methodology about adsorption experiments is somewhat confusing and not according to the results presented in the subsequent sections
9. Table 4 if both values of entropy and enthalpy changes are positive then how Gibs free energy could be negative
10. Conclusion section too long reduce its size
11. Use uniform journal style for references cited
Reviewer 2 Report
The paper described selective adsorption of palladium from metallurgical wastewater by novel thiadiazole-based chloromethyl polystyrene-modified adsorbent, viz. PS-DMTD. The FT-IR, XPS, and single-crystal X-ray diffraction methods and the density functional theory (DFT) were applied to prove the mechanism of Pd adsorption. The paper is well written and the information is given in the logical order. The paper taking into account many aspects during adsorption and the paper described interesting approaches to Pd(II) recovery.
My recommendation is Minor revision.
Detailed comments are set out below:
1. Line 98 – Please use the capital Letter at the beginning.
2. Line 114 - PS-MTD was also synthesized as described above. I can not see the description in Subchapter 2.3. for this sorbent. If this sorbent preparation was described in ref. 29 (Zhang, Q. H.; Hou, B. S.; Zhang, G. A., Inhibitive and adsorption behavior of thiadiazole derivatives on carbon steel corrosion in CO2-saturated oilfield produced water: Effect of substituent group on efficiency. J. Colloid Interface Sci 2020, 572, 91-106, 538, https://doi.org/10.1016/j.jcis.2020.03.065) this information should be highlighted. There was mentioned that MTD preparation was described in 29, what about PS-MTD? Moreover in the introduction – at the and the authors mentioned that “In this paper, two chloromethyl polystyrene resin-modified adsorbents, PS-MTD and PS-DMTD, with different microstructures, were prepared but in the paper only results for PS-DMTD could be found. The authors should add information why one of the adsorbents was rejected and based on what results. In Table S1 are schemes for two adsorbents. Please add the explanation or focus your attention only on PS-DMTD.
3. Table S1 – Based on information included in line 114: The synthesis methods of DMTD and MTD were taken from the literature [29] in Table S1 the references to DMTD and MTD synthesis should be added.
4. Line 126 – in the explanation of parameters included in Eqs. 1-4 m (g) was explained as: the mass of PS-DMTD. What about PS-MTD?
5. Line 149 – should be CH2-Cl - 2 should be in subscript.
6. Line 217-218 – Eq. 7 is a repetition of Eq. 6 – Eq. 7 should be deleted. Please renumber the equations.
7. Table S3 – Please use Pd(II) or Pd2+ for all components of wastewaters and I the paper (e.g. line 256).
8. Line 255 – please add the symbol of adsorption efficiency in parentheses to be sure that in Fig. 5 the same parameters were presented.
9. Please check line 280.

Reviewer 3 Report
This is an interesting work in which two kinds of chloromethyl polystyrene resin-modified adsorbents were prepared and used for adsorption of metal ions from metallurgical waste water. The results indicated selective adsorption for Pd(II), and the structural and batch sorption analyses provided a proper picture on the adsorption properties of the materials.
The paper is written in a sufficiently high level and could be recommended for publication, after some improvements as listed below.
Major remarks.
The batch adsorption isotherms have been displayed in linearized form and hidden in the supplementary materials. In my view, these data are of higher importance and could be presented in the main text. However, a problem is seen here: in the linearized form, both Langmuir and Freundlich isotherms show very much linear behavior. The minor difference in the R2 factor, 0.9926 vs. 0.9916, is by no ways decisive to conclude about the prevalence of one of the models. These results mean, that the sorption conditions were chosen in a too narrow sorbate concentration range, or other sorbent concentrations should have been used.
1. These present results do not allow to discriminate between the monolayer and multilayer adsorption, therefore these conclusions must be revised in the whole paper.
2. The isotherms should be measured with optimized concentrations.
3. The isotherms should be plotted in their original, non-linear form, together with the fitted Langmuir, Freundlich or other relevant models. Then the reader has the chance to see, which model fits better.
Minor remarks.
1. In Figure S11, S12, the number of experimental points is different. Please plot all data.
2. In Figure S8, the pH would be more informative than the acid concentration. Please add a pH scale, if possible.
3. As an efficient inorganic sorbent for Palladium, a recent study can be mentioned in the introduction, https://doi.org/10.1080/1539445X.2021.1999270
4. Please revise references, unify the use of fullstops in journal name abbreviations.
Round 2
Reviewer 1 Report
ok
Reviewer 3 Report
In the revised version, the problematic issues have been properly revised.
According to the new analysis, and the clear superiority of the Langmuir isotherm fitting, it can be mentioned safely, that the adsorption sites form a monolayer.